# *ATP2A2* SINE Insertion in an Irish Terrier with Darier Disease and Associated Infundibular Cyst Formation

**DOI:** 10.3390/genes11050481

**Published:** 2020-04-28

**Authors:** Monika Linek, Maren Doelle, Tosso Leeb, Anina Bauer, Fabienne Leuthard, Jan Henkel, Danika Bannasch, Vidhya Jagannathan, Monika M. Welle

**Affiliations:** 1AniCura Tierärztliche Spezialisten, 22043 Hamburg, Germany; monikalinek@gmail.com (M.L.); marendoelle@gmx.net (M.D.); 2Institute of Genetics, Vetsuisse Faculty, University of Bern, 3001 Bern, Switzerland; anina.bauer@hotmail.com (A.B.); Fabienne.Leuthard@vetmeduni.ac.at (F.L.); jan.henkel@vetsuisse.unibe.ch (J.H.); dlbannasch@ucdavis.edu (D.B.); vidhya.jagannathan@vetsuisse.unibe.ch (V.J.); 3Dermfocus, University of Bern, 3001 Bern, Switzerland; monika.welle@vetsuisse.unibe.ch; 4Department of Population Health and Reproduction, School of Veterinary Medicine, University of California, Davis, CA 95616, USA; 5Institute of Animal Pathology, Vetsuisse Faculty, University of Bern, 3001 Bern, Switzerland

**Keywords:** *Canis lupus familiaris*, dog, dermatology, skin, desmosome, acantholysis, calcium, animal model, veterinary medicine, precision medicine

## Abstract

A 4-month-old female Irish Terrier presented with a well demarcated ulcerative and crusting lesion in the right ear canal. Histological analysis revealed epidermal hyperplasia with severe acantholysis affecting all suprabasal layers of the epidermis, which prompted a presumptive diagnosis of canine Darier disease. The lesion was successfully treated by repeated laser ablation of the affected epidermis. Over the course of three years, the dog additionally developed three dermal nodules of up to 4 cm in diameter that were excised and healed without complications. Histology of the excised tissue revealed multiple infundibular cysts extending from the upper dermis to the subcutis. The cysts were lined by squamous epithelium, which presented with abundant acantholysis of suprabasal keratinocytes. Infundibular cysts represent a novel finding not previously reported in Darier patients. Whole genome sequencing of the affected dog was performed, and the functional candidate genes for Darier disease (*ATP2A2*) and Hailey-Hailey disease (*ATP2C1*) were investigated. The analysis revealed a heterozygous SINE insertion into the *ATP2A2* gene, at the end of intron 14, close to the boundary of exon 15. Analysis of the *ATP2A2* mRNA from skin of the affected dog demonstrated a splicing defect and marked allelic imbalance, suggesting nonsense-mediated decay of the resulting aberrant transcripts. As Darier disease in humans is caused by haploinsufficiency of *ATP2A2*, our genetic findings are in agreement with the clinical and histopathological data and support the diagnosis of canine Darier disease.

## 1. Introduction

The skin is the largest organ of the human body and forms an essential barrier to protect the body from fluid loss and harmful agents of the environment. The epidermis representing the outermost layer of the skin consists of a stratified epithelium with keratinocytes as its major cell type. Keratinocytes proliferate in the basal layer and subsequently undergo a highly coordinated differentiation program while they move upwards through the spinous and granular layers until they finally reach the stratum corneum, from which they are continuously shed [1]. The barrier function of the epidermis requires tight adhesion between keratinocytes, which is mainly mediated by desmosomes. Ca^2+^ signaling is essential for epidermal differentiation and intraepidermal cohesion [2,3,4,5]. Several inherited disorders of the skin involving variants in calcium pumps have been recognized [6].

In humans, Darier disease (MIM #124200), also called Darier-White disease or keratosis follicularis, is inherited as an autosomal dominant trait and caused by heterozygous variants in the *ATP2A2* gene encoding the endoplasmic/sarcoplasmic reticulum Ca^2+^-ATPase 2 (SERCA2) [7,8]. Darier disease typically starts before the third decade and is clinically characterized by warty papules and plaques in seborrheic areas (central trunk, flexures, scalp, and forehead), palmoplantar pits, and distinctive nail abnormalities [7,9]. Secondary infection is common. Neuropsychiatric abnormalities have been described in a small fraction of the patients with Darier disease [9].

Hailey-Hailey disease (OMIM #169600), also called benign chronic pemphigus, is another autosomal dominant skin disorder caused by heterozygous variants in the *ATP2C1* gene encoding a Ca^2+^-ATPase expressed in the membrane of the Golgi apparatus [10]. Hailey-Hailey disease usually becomes manifest in the third or fourth decade of life with erythema, vesicles, and painful erosions involving the body folds, particularly the groin and axillary regions [11]. Both diseases are characterized histologically by the breakdown of intercellular contacts between suprabasal keratinocytes (acantholysis) with variable dyskeratosis. Differential diagnosis is based on the skin lesion types, their distribution on the body, and subtle histological differences [9,11,12].

Many independent genetic variants in *ATP2A2* and *ATP2C1* in human patients with Darier disease or Hailey-Hailey disease have been described. Variations in the clinical and histological phenotypes may at least partly correlate with the different specific genetic variants [8]. Nonetheless, both diseases are inherited as autosomal dominant traits and are due to haploinsufficiency of the encoded calcium pumps [6].

Dermatoses affecting desmosomes in domestic animals have been summarized in a comprehensive review [13]. In one report, clinical, histological, immunohistological, and ultrastructural findings in a male English Setter and two of its female offspring were initially reported as Hailey-Hailey disease [14,15]. A subsequent study found depletion of the ATP2A2-gated stores in cultured keratinocytes from one of these dogs and suggested that these dogs had Darier disease and not Hailey-Hailey disease as previously reported [16]. To the best of our knowledge, the underlying causative genetic variant was not reported in these cases and no further cases in dogs have been reported in the scientific literature.

In the present study, we describe the clinical and histological phenotype and the genetic analysis of an Irish Terrier, which all together enabled the diagnosis of canine Darier disease. In addition to the epidermal lesions, this dog presented with multifocal infundibular cysts with suprabasal acantholysis, a feature that has never been described with Darier disease, neither in humans nor in dogs. The successful management of the skin lesions with repeated diode laser ablation is outlined.

## 2. Materials and Methods

### 2.1. Ethics Statement

All dogs in this study were privately owned and samples were collected with the consent of their owners. The collection of blood samples from control dogs was approved by the “Cantonal Committee For Animal Experiments” (Canton of Bern; permit 75/16; Approval date: 11 July 2016). The collection of samples from the affected dog was performed for diagnostic or therapeutic reasons and did not constitute an animal experiment in the legal sense.

### 2.2. Clinical Examinations and Management

A 4-month-old, intact female Irish Terrier with 9.5 kg body weight was initially presented with skin lesions in the outer ear canal in August 2016. The dog was clinically monitored for general growth, general health and skin lesion development over a period of three years. Cytology swabs were taken from crusting lesions and fine needle aspirates from nodules, respectively. Skin lesions in the outer ear canal were visualized and punch biopsies were taken via video otoscopy (Tele Pack Vet X Led, Carl Zeiss, Germany, Tuttlingen) under general anesthesia with endotracheal intubation. Nodules were excised in toto. Tissue samples for histological evaluation were fixed in 10% buffered formalin immediately.

Blood was taken twice for complete blood count and genetic testing. Tear production was assessed by Schirmer’s tear test, as recommended during vitamin A therapy. For laser ablation, an MLT Type 109 classic diode laser (Medizinische Laser Technologie GmbH, Ingelheim, Germany) with 4.0 W wave mode in continuous contact mode was used.

### 2.3. Histopathology

Biopsies were evaluated from the two plaque-like, partially eroded lesions of the external ear canal, one nodular lesion from the hind leg, and two large nodules from the neck. Tissue was processed routinely and stained with hematoxylin and eosin.

### 2.4. Whole Genome Sequencing

Genomic DNA was isolated from EDTA blood of the affected dog with the Maxwell RSC Whole Blood Kit using a Maxwell RSC 48 instrument (Promega, Madison, WI, USA). An Illumina TruSeq PCR-free DNA library (Illumina, San Diego, CA, USA) with ~350 bp insert size of the affected dog (IT390) was prepared. We collected 269 million 2 × 150 bp paired-end reads on a HiSeq 3000 instrument (32 × coverage). Mapping and alignment were performed as described [17]. The sequence data were deposited under the study accession PRJEB16012 and the sample accession SAMEA104283467 at the European Nucleotide Archive.

### 2.5. Variant Calling

Variant calling was performed using the Genome Analysis Toolkit (GATK) HaplotypeCaller [18] in gVCF mode as described [17]. To predict the functional effects of the called variants, SnpEff [19] software, together with NCBI annotation release 105 for the CanFam 3.1 genome reference assembly, was used. For variant filtering we used 655 control genomes, which were either publicly available [20,21] or produced during other projects of our group [17] (Appendix A). Structural variants were identified by visual inspection of the Illumina short read alignments in the Integrated Genome Viewer (IGV) [22]. The genotypes at the *ATP2A2* SINE insertion (Chr26:8,200,944_8,200,945ins205) were also derived by visual inspection of the short read alignments in IGV. Samples were genotyped as homozygous ref/ref, if they did not show any signs of a structural variant at this position and had at least 4 reads aligning from Chr26:8,200,929-8,200,945, thus spanning the 15 nucleotide duplication at the insertion site.

### 2.6. Gene Analysis

We used the CanFam 3.1 dog reference genome assembly and NCBI annotation release 105. Numbering within the canine *ATP2A2* gene corresponds to the NCBI RefSeq accession numbers NM_001003214.1 (mRNA) and NP_001003214.1 (protein).

### 2.7. RT-PCR and Sanger Sequencing

Total RNA was extracted from skin tissues using the RNeasy mini kit (Qiagen, Hilden, Germany). The tissue was first finely crushed in TRIZOL (Thermo Fisher Scientific, Waltham, MA, USA) using mechanical means, chloroform was then added and the RNA was separated by centrifugation. The RNA was cleared of genomic DNA contamination using the Quantitect Reverse Transcription Kit (Qiagen). The same kit was used to synthetize cDNA, as described by the manufacturer. RT-PCR was carried out using primer ATP2A2_Ex14_F, TCCTCCAAGGATTGAAGTGG, located in exon 14 and primer ATP2A2_Ex16_R, TGTCACCAGATTGACCCAGA, located in exon 16 of the *ATP2A2* gene.

After treatment with exonuclease I and alkaline phosphatase, cDNA amplicons were sequenced on an ABI 3730 DNA Analyzer (Thermo Fisher Scientific) using the forward primer ATP2A2_Ex14_F as sequencing primer. Sanger sequences were analyzed using the Sequencher 5.1 software (GeneCodes, Ann Arbor, MI, USA).

PCR on genomic DNA was performed using AmpliTaqGold360Mastermix (Thermo Fisher Scientific) and primers ATP2A2_Ex14_F (same as above) and ATP2A2_Ex15_R, TCAGGGCAGGAGCATCATTC. Genomic PCR products were also sequenced using the forward primer ATP2A2_Ex14_F as sequencing primer.

### 2.8. Whole Transcriptome Analysis (RNA-seq)

RNA libraries were prepared from total RNA of lesional and non-lesional skin of the affected Irish Terrier using the Illumina TruSeq Stranded mRNA Library Kit according to the manufacturer’s instructions. The libraries were sequenced with 2 × 50 bp paired-end sequencing chemistry on an Illumina NovaSeq 6000 instrument. The reads were mapped with STAR aligner version 2.6.0 [23] to the CanFam3.1 reference genome assembly. The sequence data were deposited under the study accession PRJEB33508 and sample accessions SAMEA6800286 and SAMEA6800287 at the European Nucleotide Archive. The read alignments of the affected Irish Terrier were visually compared to a skin RNA-seq dataset from a healthy control dog (ENA project accession PRJEB33508, sample accession SAMEA6800283).

## 3. Results

### 3.1. Clinical Examination and Management

During the first consult in August 2016, a 4-month old intact female Irish Terrier presented with several confluent, well demarcated, proliferative, crusted, and partially eroded to ulcerated plaques at the concave pinnae of the right ear extending into the medial aspect of the vertical ear canal (Figure 1A). These lesions had been present since at least 4 weeks prior to the examination. Culture swabs taken by the referring veterinarian revealed *Staphylococcus pseudintermedius* sensible to most antibiotics. The presence of a foreign body had been excluded by an ear flush. At the time of presentation, the dog received amoxicillin/clavulanic acid at a dosage of 26 mg/kg body weight (BW) twice daily (Synulox®, Zoetis, Berlin, Germany) and prednisolone 5 mg/kg BW once daily (Prednisolon 5 mg®, CP Pharma mbH, Burgdorf, Germany). Several commercially available ear cleansers and eardrops had been applied before without any improvement.

The general and dermatological examination did not reveal any abnormalities except the moderately painful and mildly pruritic lesions of the right pinna and ear canal. Cytology showed clusters of acantholytic keratinocytes, non-degenerated neutrophils, and numerous cocci.

Video otoscopy showed intact eardrums and normal horizontal ear canals in both ears. The medial aspect of the right vertical ear canal revealed well demarcated, ulcerated lesions covered with thick crusts confluent with the lesions of the concave pinna (Figure 1B). Waiting for the biopsy results, the dog was treated with squalene ear cleanser every other day, twice daily topically with Triamcinolone Acetonide cream (Volon® A Haftsalbe 1 mg/g, Dermapharm AG) and sulfadiazine creme (Flammizine®, Alliance Pharmaceuticals Limited, Chippenham, UK), changed to customized eardrops of 1% fluoroquinolone in saline solution (Baytril® 5%, Bayer AnimalHealth GmbH, Leverkusen, Germany) for easier handling. After the preliminary diagnosis of canine Darier disease, treatment with vitamin A 10,000 IU (Vitamin-A-saar®, Cephasaar, Ingbert, Germany) orally for two weeks daily; then, every other day was initiated and maintained for 3 months.

As no improvement was noticed after 3 months, we decided to ablate the lesional epidermis with a diode laser to remove the defect skin and provoke secondary healing from the periphery. This procedure was partially successful the first time. All lesions healed without crusts after repeated laser ablation of the affected tissue another three times, two, five, and 12 months apart (Figure 1C). On the pinna, small nodules of 1–2 mm remained. They were clinically and cytologically diagnosed as comedones.

In the following three years after the first presentation, the dog developed three well-demarcated, dermal nodules ranging from 2.5 cm to 4 cm in diameter on the dorsal neck, the left side of the neck, and on the right knee. Fine needle aspirates of all nodules revealed clusters of nucleated round to oval keratinocytes with mild anisocytosis and anisocaryosis and two to three nucleoli in the nucleus. These nodules were fully excised and submitted for histopathology. At the time of writing, no further lesions or nodules had developed.

### 3.2. Histopathology

Biopsies from the external ear canal revealed focally extensive epidermal hyperplasia with severe acantholysis affecting all suprabasal layers of the epidermis and resulting in in the formation of multiple small clefts and lacunae (Figure 2). Acantholytic keratinocytes were frequently dyskeratotic forming “corps ronds” (e.g., round bodies characterized by small pyknotic nuclei, a perinuclear clear halo and eosinophilic cytoplasm) or “grains” (cells with elongated nuclei present mainly in the stratum corneum and the granular layer). The epidermis was covered by compact orthokeratotic or parakeratotic keratin intermingled with dyskeratotic acantholytic cells. In areas of abundant acantholyisis, keratin extended as prominent focal plugs into the epidermis.

In all biopsies from haired skin, one or multiple infundibular cysts measuring between 0.8 × 0.5 × 0.5 cm up to 3.5 × 3.0 × 1.2 cm were extending from the upper dermis to the subcutis. The cysts were lined by squamous epithelium, which presented with abundant acantholysis of suprabasal keratinocytes. The cysts were filled with parakeratotic keratin and numerous acantholytic and dyskeratotic cells. In one biopsy from the neck, the epidermis overlying the cyst presented with severe hyperplasia and suprabasal acantholysis comparable to the findings described for the outer ear canal. Similar findings were also present in the infundibular epithelium of some hair follicles.

### 3.3. Identification of a Candidate Causative Variant

We sequenced the genome of the affected dog at 32 × coverage and called single nucleotide and small indel variants with respect to the reference genome. The variants were compared to whole genome sequence data of 8 wolves and 647 control dogs from genetically diverse breeds and searched for private protein-changing variants in the two functional candidate genes *ATP2A2* and *ATP2C1*. This analysis of small variants did not identify any likely candidate causative variants for the phenotype (Appendix A).

We then visually inspected the short read alignments in *ATP2A2* and *ATP2C1* to search for structural variants that would have been missed by our automated variant detection pipeline. Several truncated read alignments at the end of intron 14 of the *ATP2A2* gene indicated a potential insertion event including the duplication of 15 nucleotides flanking the insertion site. The inserted sequence represented a tRNA derived SINE (Figure 3A,B).

The genotypes at the SINE insertion site were investigated in the 655 control genomes. A total of 592 genomes had at least four reads spanning the insertion site and were genotyped as homozygous wildtype. In the remaining 63 control genomes, we did not see any indication for an insertion event. However, due to low coverage and/or short read lengths, the genotypes in these samples could not be reliably determined (Appendix A). PCR amplification with flanking primers on a genomic DNA sample from the affected dog provided independent confirmation of the presence of a ~205 bp insertion in heterozygous state (Figure 3C).

### 3.4. Analysis of the ATP2A2 mRNA

We next investigated whether the SINE insertion into intron 14 had any effect on the expressed *ATP2A2* mRNA. Initial RT-PCR experiments on RNA from skin of the affected dog with different primer combinations yielded products of the expected size and sequence and did not indicate any obvious qualitative defects in mRNA splicing.

As the genomic insertion was only present in a heterozygous state, the wildtype allele was still expected to give rise to the normal *ATP2A2* transcript. Consequently, a potential splicing defect leading to nonsense mediated decay in transcripts from the mutant allele or transcriptional silencing of the mutant allele would not have been detected by our qualitative analysis of RT-PCR bands. We therefore additionally investigated the allele-specific expression of *ATP2A2* transcripts. This analysis demonstrated that ~85% of the detected transcripts were derived from the wildtype allele with an almost complete absence of transcripts from the mutant allele (Figure 4A,B).

To gain further insights into possible splicing defects, we performed an RNA-seq experiment and whole transcriptome analysis in skin of the affected dog. Visual inspection of the short-read alignments in the region of the *ATP2A2* gene confirmed the strong allelic bias of the transcripts. Furthermore, this experiment revealed the presence of rare transcripts containing an additional, aberrantly spliced exon. This 139 nt exon was derived from genomic sequence a short distance upstream of the SINE insertion (Chr26:8,200,774-8,200,912). The aberrant exon contained an early premature stop codon. The variant designation of the predicted protein from transcripts containing the aberrant exon is NP_001003214.1:p.(Thr700Valfs*6). Only a small proportion of the transcripts from the mutant allele was correctly spliced and had the correct coding sequence (Figure 4C and Appendix A).

## 4. Discussion

In this study, we describe the clinical and histological phenotype and the genetic analysis of an Irish Terrier with canine Darier disease. The dog developed two different types of clinical lesions over a follow up time of 3 years. One lesion type presented as demarcated, proliferative, crusted and eroded to ulcerated and was present on the right concave pinna and in the ear canal. This lesion type was overlying an infundibular cyst, which represented the second type of lesion.

The crusted lesions were more severe, painful, and pruritic than the lesions described in the previously published cases [14,15]. Essentially the published cases describe one seven-month-old, male, intact English Setter that exhibited a peculiar crusting lesion on the ventral chest and two of his six living offspring that were intentionally bred by mating the affected English Setter to a normal laboratory Beagle. The two Setter-Beagle crossbred dogs developed similar lesions as the sire with alopecia, erythema, and hyperplasia on the lateral knee (one dog) or dorsal head (second dog) at the age of four and seven weeks, respectively. The lesions slightly enlarged and worsened during adolescence but remained static thereafter and did not require therapy. The histopathology and ultrastructural findings were similar in all three dogs and initially considered to represent Hailey-Hailey disease (benign familiar chronic pemphigus) [14,15]. In a subsequent study, cultured keratinocytes from one of these dogs were investigated and a depletion of ATP2A2-gated Ca^2+^ stores was found. This finding suggested that these dogs had Darier disease rather than Hailey-Hailey disease [16].

Considering the clinical presentation of focal hyperplastic skin lesions in these dogs [14,15], their early age of onset, and the histology with severe acantholysis with prominent dyskeratosis and the formation of corps ronds and grains also suggests that they had Darier disease and not Hailey-Hailey disease as previously reported. In Hailey-Hailey disease, prominent suprabasal acantholyisis is also a feature, but loss of keratinocyte cohesion is not as complete as in Darier disease and detached keratinocytes still form clusters. Dyskeratosis is milder than in Darier disease [12].

The specific molecular mechanisms that lead to the multifocal hyperproliferation, dyskeratosis and acantholysis of epidermal keratinocytes have not yet been identified. It is well known that extracellular calcium plays a crucial role in regulating differentiation and adhesion of cultured keratinocytes [6,16]. Low levels of Ca^2+^ induce keratinocyte proliferation while physiological levels induce cell-to-cell adhesion and keratinocyte differentiation. It has been shown that changes in the intracellular calcium homeostasis in Darier disease impair processing, transport, and assembly of calcium-dependent desmosomal proteins. Desmosomes provide strong adhesive bonds between neighboring cells by correct assembly of their intercellular and intracellular proteins and disturbance of this process results in acantholysis [6,13,16]. An alternative hypothesis is that the defective calcium homeostasis leads to delayed exit of keratinocytes from the cell cycle, which may promote secondary mutations that lead to acantholysis [16]. Furthermore, Ca^2+^ levels in the endoplasmic reticulum play a key role in post-translational modification of proteins. Disturbances of Ca^2+^ homeostasis may result in an accumulation of unfolded proteins in the endoplasmic reticulum and subsequent apoptosis. It has been suggested that the “corps ronds” in Darier disease may be the result of such an impaired protein folding [6].

In our patient, the crusting lesions did not enlarge, and only one similar lesion developed on other parts of the body. However, over a course of 3 years, the dog developed three infundibular cysts where the Darier specific acantholysis with dyskeratosis was seen within the cyst walls. In the epidermis overlying one cyst on the neck, a similar epidermal lesion as described for the ear developed. To the best of our knowledge, infundibular cysts have not yet been described in humans or dogs with Darier disease. In humans, several clinical variants of Darier disease including vesicobullous, hypopigmented, cornifying, zosteriform or linear, acute, and comedonal subtypes have been described [9]. Comedonal Darier disease is a very rare variant with severe follicular involvement and characterized by open or closed comedones with central keratotic plugs and the presence of greatly elongated epidermal protrusions at the base of the comedones [24,25]. However, in the canine case presented here, no comedones but true infundibular cysts without the described elongated papillary projections at the base of the cysts were present. Thus, this presentation of our case is new and has never been described in humans or dogs.

The dermal nodules were successfully excised and no recurrence was noted at the site of excision. The hyperplastic and ulcerated lesions on the pinna and the ear canal required treatment, as they were painful and prone to secondary infection at any time-point.

In human medicine, numerous therapeutic options have been described, including systemic or topical retinoids, cyclosporine, vitamin A, systemic or topical corticosteroids, topical 5-fluorouracil, keratolytics with urea, or interventional treatment, like dermabrasion, laser ablation, and excision [26,27,28,29,30].

The age of the dog, the difficulty in the application of topical treatments, and financial and psychological restrictions of the owner limited the treatment options in our patient. Systemic and topical glucocorticoids, as well as systemic vitamin A, were not successful. As other medical treatments were denied, we treated the lesions in the ear with a diode laser. Carbon dioxide (CO_2_) laser or yttrium aluminum garnet (YAG) laser ablation in Darier disease and Hailey- Hailey disease has been reported as a successful treatment option in human medicine [28,29,30]. We chose a diode laser, as this device can be used via the working channel of the video otoscope and allowed us to ablate the lesional skin of the medial aspect of the vertical ear canal under visual control. The concept of laser therapy in Darier disease is the ablation of the defective epidermis and the follicular infundibulum, which might be the focus of recurrence. In the described laser-treated human patients, remodeling of normal skin, as well as cicatrization, occurred. In our canine patient, the treatment was very well tolerated and led to full resolution of the lesion after several interventions. The ear canal tissue provides only a thin layer of dermis over the underlying cartilage, which is prone to necrosis if damaged. Therefore, our inventions had most likely not been aggressive enough to completely destroy the affected tissue and the follicular infundibulum in one session. In less vulnerable areas of the skin, a laser treatment might be more favorable.

Darier disease in human patients is caused by heterozygous genetic variants in *ATP2A2*. These include missense, nonsense, frameshift, and splice site variants [7,8,31]. An intronic 18 bp insertion, 12 nucleotides upstream of exon 3, caused Darier disease in one human family. This insertion altered splicing and resulted in an aberrant transcript with 6 additional codons, which could be detected as in-frame insertion that did not lead to nonsense mediated decay [31].

Our genetic analysis revealed a heterozygous SINE insertion in intron 14 of the *ATP2A2* gene, which was exclusively found in the affected dog, but not in 592 controls. The functional analysis at the mRNA level indicated nearly mono-allelic expression of *ATP2A2* transcripts in skin of the affected dog. RNA-seq showed that the SINE insertion led to the activation of cryptic splice sites in intron 14 and the inclusion of an aberrant exon containing a premature stop codon. The observed allelic imbalance of the transcripts can be plausibly explained by nonsense-mediated decay [32] of the transcripts with the premature stop codon.

A limitation of our genetic analysis is the lack of family data. We hypothesize that the SINE insertion in the affected dog is the consequence of a *de novo* mutation event. Thus, the parents of the dog are assumed to be phenotypically and genotypically wildtype. If the homozygous wildtype genotype were confirmed in both parents, this would provide proof for the hypothetical *de novo* mutation event and another strong supporting argument for the pathogenicity of the SINE insertion. Unfortunately, we did not have access to the parents of the affected dog.

Given the extensive functional knowledge on *ATP2A2* and the role of *ATP2A2* variants in human Darier disease, we nonetheless think that our data strongly suggest that the SINE insertion may be considered a candidate causative variant for the phenotype in the affected dog.

## 5. Conclusions

We provided a comprehensive clinical, histopathological and genetic characterization of an Irish Terrier with Darier disease. The genetic analysis revealed an intronic SINE insertion into *ATP2A2* as a candidate causative genetic variant leading to aberrant splicing and degradation of aberrant transcripts.

## Figures and Tables

**Figure 1 genes-11-00481-f001:**
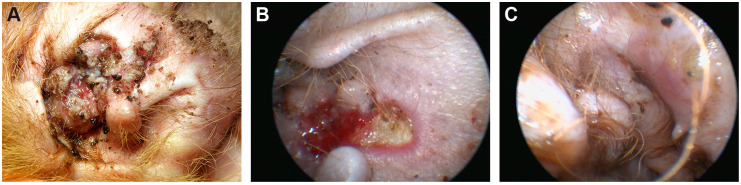
Clinical phenotype. (**A**) Concave pinnae of the right ear showing well demarcated crusting, eroded and ulcerated skin plaques. (**B**) Medial aspect of the right ear canal with well demarcated ulcerated lesions visualized via video otoscopy after crusts had been flushed away. (**C**) Same aspect of the ear canal: Intact, slightly erythematous skin after repeated laser ablation.

**Figure 2 genes-11-00481-f002:**
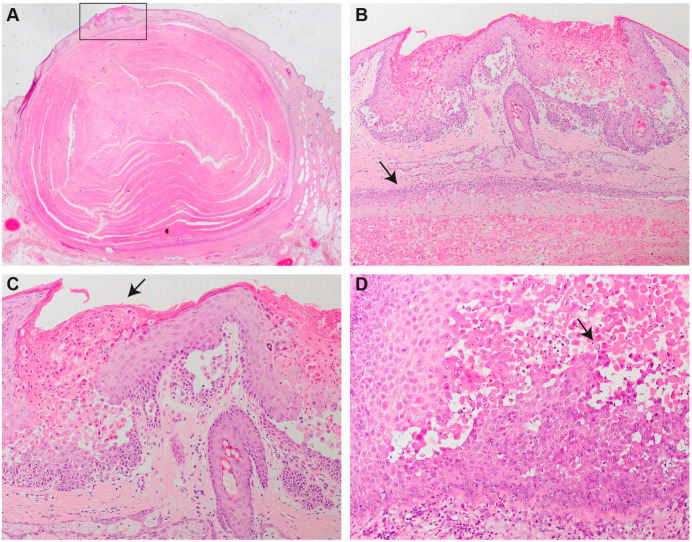
Histopathology. (**A**) Infundibular cyst underneath a focal area of hyperplastic epidermis with abundant suprabasal acantholyisis (rectangle). (**B**) Higher magnification of the focal area of hyperplastic epidermis with suprabasal acantholysis overlying the cyst wall (arrow). The epidermal plaque is characterized by severely irregularly hyperplastic epidermis with abundant suprabasal acantholytic and dyskeratotic keratinocytes forming the “corps ronds” typical for Darier disease. Keratotic plugs composed of parakeratotic keratin and grains extend into the clefts resulting from abundant acantholysis. The hyperplastic plaque is overlying an infundibular cyst composed of squamous epithelium with abundant suprabasal acantholysis. The cyst is filled with parakeratotic keratin and numerous “corps ronds”. (**C**) Higher magnification of the lesions already presented in (**A**,**B**). Note the abundant suprabasal acantholyis of dyskeratotic keratinocytes forming “corps ronds”, “grains” and parakeratotic keratin (arrow). (**D**) Hyperplastic plaque from the outer ear canal. Within the severely hyperplastic epidermis, numerous acantholytic and dyskeratotic keratinocytes forming “corps ronds” (arrow) and causing small clefts are present.

**Figure 3 genes-11-00481-f003:**
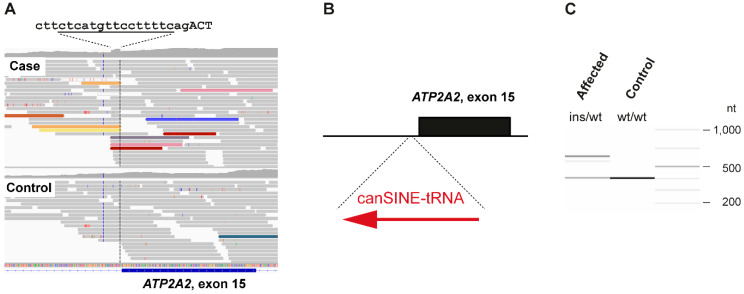
SINE insertion into intron 14 of the *ATP2A2* gene. (**A**) IGV screenshot illustrating the structural variant. The case shows an increased coverage over 15 nucleotides spanning from Chr26:8,200,930-8,200,944 (CanFam3.1 assembly). The sequence at the intron/exon boundary is given with the duplicated bases underlined. Capital letters represent the first 3 bases of exon 15. Several read alignments are soft-clipped at the left or right boundary of the duplicated 15 nt region. Colored reads indicate that their mates map to other chromosomes. These features are characteristic for an insertion of a repetitive element into the genome of the affected dog. (**B**) Schematic representation of the SINE insertion. A ~205 bp canine SINE-tRNA insertion was found in heterozygous state in the affected Irish Terrier. (**C**) Experimental genotyping of the SINE insertion by fragment size analysis. We amplified the intron 14/exon 15 boundary of the *ATP2A2* gene by PCR in the affected dog and a control and separated the products by capillary gel electrophoresis.

**Figure 4 genes-11-00481-f004:**
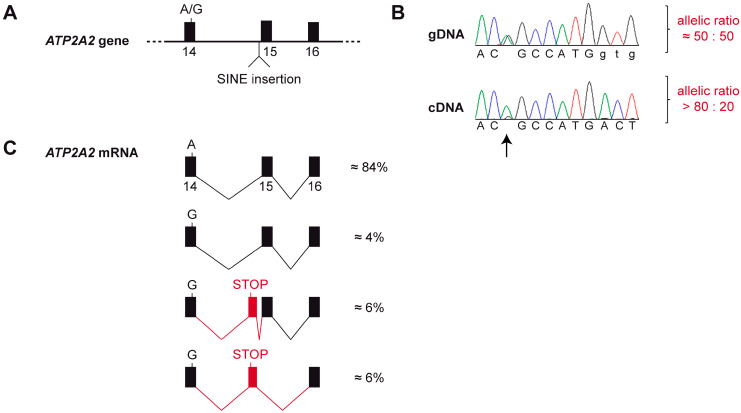
Splicing defect and allele-specific expression of the *ATP2A2* mRNA. (**A**) Genomic organization of the *ATP2A2* gene. The affected dog was heterozygous at the silent c.2091A>G variant located at the end of exon 14 and heterozygous for the SINE insertion in intron 14. (**B**) A Sanger electropherogram obtained from a genomic PCR product shows the expected equal ratio of the two alleles at c.2091A>G. In contrast, a Sanger electropherogram obtained with the same sequencing primer from a cDNA amplicon showed a strong bias towards the A-allele (arrow). This semi-quantitative analysis suggests that the transcripts from the mutant *ATP2A2* allele are degraded, possibly by nonsense mediated decay or another mechanism of the cellular quality control. (**C**) RNA-seq analysis from skin of the affected dog confirmed the strong allelic bias in the transcripts. Only very little functional transcripts are produced from the G-allele. The majority of the transcripts from the G-allele contain an aberrant exon and a premature stop codon. Further details of the RNA-seq analysis are shown in Appendix A.

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
