# Peer review of "ATP2A2 SINE Insertion in an Irish Terrier with Darier Disease and Associated Infundibular Cyst Formation"

_genes, 2020, doi:10.3390/genes11050481_

Round 1
Reviewer 1 Report
The manuscript by Linek M, et al. regarding Darier disease in an Irish terrier is a well-written and well-supported clinical case report with molecular diagnosis using whole genome sequencing and a candidate gene approach. The authors claim also that the pathology in this dogs constitutes a new form of the disease in humans or dogs.
Introduction: This section is important because the disease will be unfamiliar to many readers. Is it true that there are no pertinent references in the last decade?
Methods: The methods are routine and described in sufficient detail.
Results: The final result suggesting haploinsufficiency of the <i>ATP2A2<i> mRNA is pretty thin. Because the peak area analysis is only semi-quantitative, and in our experience artifacts often masquerade as heterozygous SNPs, it would be better to demonstrate more than one heterozygous SNP in the mRNA that showed similar allelic disproportion. It would be best if at least one of them is in exon 15 or 16, somewhere on the opposite side of the insertion from the one shown.
The second issue is that a lot of emphasis is placed on NMD of the mRNA from the inserted chromosome. NMD would depend on creation of an early stop codon. Is there any direct evidence of exon skipping? If not, another explanation should be discussed. I don't believe that financial constraint of the molecular laboratory should prevent attempts to clone the molecules of cDNA representing the normal and mutant mRNAs.
Discussion: The emphasis of the discussion is on the clinical and pathological aspects of the case as well as the candidate-gene approach to molecular diagnosis. What is missing here is the pathophysiology bridge; there needs to be some discussion of how the lack of SERCA calcium stores creates the observed acantholysis.
Minor suggestions:
Lines 71 and 281, "consecutive" is the wrong word, use "subsequent" instead.
Line 282, depletion was found, not depletion were found.
Line 284, a better first word of the sentence would be "Considering" rather than "Revisiting"
Line 315, We chose (past tense), not we choose.
Line 318, defective epidermis, not defect epidermis.
Line 330, .... in-frame insertion and did not...
Lines 335 and 342, delete the "e.g.", it is redundant following "such as"
Line 416 of reference 14, the article title has duplicate "Canine". Please also give the year in bold font.
Line 441, bold font the year.
Author Response
(1)
The manuscript by Linek M, et al. regarding Darier disease in an Irish terrier is a well-written and well-supported clinical case report with molecular diagnosis using whole genome sequencing and a candidate gene approach. The authors claim also that the pathology in this dogs constitutes a new form of the disease in humans or dogs.
Response: Thank you for the compliments!
(2)
Introduction: This section is important because the disease will be unfamiliar to many readers. Is it true that there are no pertinent references in the last decade?
Response: Yes, this is true, although the authors know that one or two unpublished cases have been seen by dermatologists / pathologists. We added the following statement to the introduction “… and no further cases in dogs have been reported in the scientific literature.” (lines 73/74).
(3)
Methods: The methods are routine and described in sufficient detail.
Response: No change requested.
(4)
Results: The final result suggesting haploinsufficiency of the ATP2A2 mRNA is pretty thin. Because the peak area analysis is only semi-quantitative, and in our experience artifacts often masquerade as heterozygous SNPs, it would be better to demonstrate more than one heterozygous SNP in the mRNA that showed similar allelic disproportion. It would be best if at least one of them is in exon 15 or 16, somewhere on the opposite side of the insertion from the one shown.
Response: We actually have RNA-seq data from the affected dog that were lost from my memory during the transition of several graduate students. We now properly analyzed the RNA-seq data and they convincingly demonstrate the inclusion of an aberrant exon with a premature stop codon in the majority of the transcripts derived from the mutant allele. The additional data are now described in the results section, in Figure 4C and in Supplementary Figure 1.
(5)
The second issue is that a lot of emphasis is placed on NMD of the mRNA from the inserted chromosome. NMD would depend on creation of an early stop codon. Is there any direct evidence of exon skipping? If not, another explanation should be discussed. I don't believe that financial constraint of the molecular laboratory should prevent attempts to clone the molecules of cDNA representing the normal and mutant mRNAs.
Response: With the new data, we provide evidence for the inclusion of an aberrant exon with a premature stop codon. This provides a plausible explanation for the NMD.
(6)
Discussion: The emphasis of the discussion is on the clinical and pathological aspects of the case as well as the candidate-gene approach to molecular diagnosis. What is missing here is the pathophysiology bridge; there needs to be some discussion of how the lack of SERCA calcium stores creates the observed acantholysis.
Response: We added a paragraph discussing the suggested underlying pathophysiology (lines 312-326).
(7)
Minor suggestions:
Lines 71 and 281, "consecutive" is the wrong word, use "subsequent" instead.
Line 282, depletion was found, not depletion were found.
Line 284, a better first word of the sentence would be "Considering" rather than "Revisiting"
Line 315, We chose (past tense), not we choose.
Line 318, defective epidermis, not defect epidermis.
Line 330, .... in-frame insertion and did not...
Lines 335 and 342, delete the "e.g.", it is redundant following "such as"
Line 416 of reference 14, the article title has duplicate "Canine". Please also give the year in bold font.
Line 441, bold font the year.
Response: Thank you very much for spotting these errors. We revised all of them accordingly.
Reviewer 2 Report
This research shows much of information about canine Darier disease with respect of gene (ATP2A2). This finding also useful in human Darier disease.
Experiment was performed completely to investigate the canine Darier disease comparing with that's of human.
English writing and description is excellent.
I want to recommend this article as "accept in present form"
Author Response
No changes required.